# Bone Formation in Zebrafish: The Significance of DAF-FM DA Staining for Nitric Oxide Detection

**DOI:** 10.3390/biom13121780

**Published:** 2023-12-12

**Authors:** Ann Huysseune, Ulrike G. Larsen, Daria Larionova, Cecilie L. Matthiesen, Steen V. Petersen, Marc Muller, P. Eckhard Witten

**Affiliations:** 1Research Group Evolutionary Developmental Biology, Biology Department, Ghent University, K.L. Ledeganckstraat 35, 9000 Ghent, Belgium; daria.larionova@ugent.be (D.L.); peckhardwitten@aol.com (P.E.W.); 2Department of Zoology, Faculty of Science, Charles University, Vinicna 7, 128 44 Prague, Czech Republic; 3Department for Biomedicine, Aarhus University, Høegh-Guldbergs Gade 10, 8000 Aarhus, Denmark; ul@biomed.au.dk (U.G.L.); cl@biomed.au.dk (C.L.M.); svp@biomed.au.dk (S.V.P.); 4Laboratoire d’Organogenèse et Régénération, GIGA-R 1, Avenue de l’Hôpital, B34 Sart Tilman, 4000 Liège, Belgium; m.muller@uliege.be

**Keywords:** zebrafish, osteoblasts, ossification, nitric oxide, notochord sheath, bulbus arteriosus

## Abstract

DAF-FM DA is widely used as a live staining compound to show the presence of nitric oxide (NO) in cells. Applying this stain to live zebrafish embryos is known to indicate early centers of bone formation, but the precise (cellular) location of the signal has hitherto not been revealed. Using sections of zebrafish embryos live-stained with DAF-FM DA, we could confirm that the fluorescent signals were predominantly located in areas of ongoing bone formation. Signals were observed in the bone and tooth matrix, in the notochord sheath, as well as in the bulbus arteriosus. Surprisingly, however, they were exclusively extracellular, even after very short staining times. Von Kossa and Alizarin red S staining to reveal mineral deposits showed that DAF-FM DA stains both the mineralized and non-mineralized bone matrix (osteoid), excluding that DAF-FM DA binds non-specifically to calcified structures. The importance of NO in bone formation by osteoblasts is nevertheless undisputed, as shown by the absence of bone structures after the inhibition of NOS enzymes that catalyze the formation of NO. In conclusion, in zebrafish skeletal biology, DAF-FM DA is appropriate to reveal bone formation in vivo, independent of mineralization of the bone matrix, but it does not demonstrate intracellular NO.

## 1. Introduction

Since its discovery in 1987 as a signaling molecule in the cardiovascular system, nitric oxide (NO) has been identified as a major player in a wide range of physiological processes, including neurotransmission and immune responses (recently reviewed in, e.g., [1,2]).

Because NO is a gaseous, free radical molecule with a short lifetime, its demonstration most often depends on localizing the enzyme that catalyzes the production of NO from L-arginine, nitric oxide synthase, or NOS. For example, in skeletal tissues, osteoblasts, osteocytes, and osteoclasts, cells that moderate aspects of bone formation and bone resorption, express different isoforms of NOS [3,4]. Furthermore, NO has been found to have a regulatory role in bone with complex actions in both osteoblastic and osteoclastic lineages [5,6,7,8,9].

Over time, methods have also been developed to directly reveal the presence of NO. Kojima et al. [10] designed several diaminofluoresceins (DAFs) including DAF-2. These DAFs are converted to the triazole form by a reaction with NO, thereby greatly enhancing their fluorescence. The diacetate ester derivative (DAF-2 DA) allows NO imaging in living cells as it easily enters the cells to be converted into DAF-2 by intracellular esterases. DAF-2 then reacts with NO to form DAF-2T (the triazole form of DAF-2), which becomes trapped in the cytosol. Kojima et al. [11] next designed DAF-FM by introducing a methyl group and replacing chlorine with fluorine atoms in DAF-4. This derivative (4-amino-5-methylamino-2′-7′-difluorofluorescein, or DAF-FM) and its diacetate ester (DAF-FM DA) have found more general usage. Its triazole form, DAF-FM T, is strongly fluorescent, less susceptible to photobleaching, and the fluorescence intensity is stable above pH 5.8 [11]. Therefore, DAF-FM can detect lower levels of NO. Like DAF-2 DA, DAF-FM DA spontaneously crosses the plasma membrane and is subsequently cleaved by esterases to generate intracellular DAF-FM. The latter is then oxidized by NO to form a triazole product. When measured in a buffer, this reaction increases the fluorescence quantum yield by about 160-fold [11].

The zebrafish, a small vertebrate that is particularly amenable to experimentation and live imaging, has been a model of choice in revealing NO in the skeletal system via live staining. Several authors have reported that DAF-FM DA, applied as live staining on embryos and early postembryonic stages, is a reliable method to label early ossification centers in developing zebrafish [12,13,14]. However, in these studies, the method was employed to detect NO on whole-mount embryos but not to detect NO at a cellular level.

Because of the growing field of biomedical research using zebrafish to develop skeletal disease models, as well as the interest of the NO field, we set out to refine these observations and to assess the precise (cellular) localization of DAF-FM DA signals in the developing zebrafish embryo. We found that DAF-FM DA stains the extracellular matrix in early centers of bone formation, independent of mineralization of the bone matrix, as well as in the notochord sheath and the bulbus arteriosus. The significance of NO for the function of osteoblasts is nevertheless revealed by the absence of bones and of DAF-FM DA staining after the inhibition of NO production.

## 2. Materials and Methods

### 2.1. Sample Collection

Adult wildtype (WT) zebrafish (*Danio rerio*) were maintained and embryos were raised under standard conditions at 28.5 °C and a 14/10 h light/dark cycle, according to [15].

### 2.2. NO Staining on Live Material

Live staining was performed both on manually dechorionated zebrafish embryos (30, 36, 42 h post-fertilization, hpf) as well as on early postembryonic stages of zebrafish up to the first independent feeding stage (5 days post-fertilization, dpf, for zebrafish [16]).

DAF-FM DA was used as a live stain according to a published protocol [12]. Briefly, specimens were soaked for up to 3 h (times as indicated) in a freshly made solution of 5 µM DAF-FM DA in 0.1% DMSO, shielded from light. The controls were simultaneously soaked in 0.1% DMSO. To test the specificity of DAF-FM DA staining for NO, staining with DAF-FM DA was also carried out after a treatment of 30 min with NO scavenger 2-(4-carboxyphenyl)-4,4,5,5-tetramethylimidazoline-1-oxyl-3-oxide sodium salt (c-PTIO, EMD Millipore, Burlington, MA, USA) at a concentration of 500 µM in 0.05% DMSO. After staining, specimens were rinsed in PBS and sedated using MS222. Unless stated otherwise, images of live, sedated animals were taken using a Zeiss Axio Zoom V16 Fluorescence Stereo Zoom microscope equipped with a 5 MP CCD camera for imaging under standard magnification (20 and 80×) and standard illumination (50,000 and 110,000 ms, resp.). Particular images were recorded using a Zeiss Axio Observer Z1 microscope equipped with an apotome 2 unit (inverted setup). After imaging, animals were euthanized using an overdose of MS222 and fixed in 4% PFA for processing in glycol methacrylate (GMA).

### 2.3. NO Staining on Fixed Material

WT zebrafish of 8 dpf were euthanized using an overdose of MS222 and fixed either in 4% paraformaldehyde (PFA), in acetone, or in 70% ethanol and exposed to 5 µM DAF-FM DA in 0.1% DMSO or to the control vehicle for 3 h in the dark. They were next immediately observed and photographed.

### 2.4. NO Staining after Inhibition of NOS

DAF-FM DA staining was also performed after the inhibition of NO production using 1-[2-(trifluoromethyl)phenyl]imidazole (TRIM, Sigma-Aldrich, St. Louis, MO, USA). TRIM specifically inhibits the function of the enzymes catalyzing the formation of NO, nitric oxide synthases (NOS), in particular nNOS and iNOS, and eNOS with a lower efficiency. The latter is, however, not present in zebrafish [17]. TRIM was used as a 100 µM solution in the embryo medium, prepared from a 10 mM stock solution in 10% DMSO, and applied in darkness uninterruptedly on WT zebrafish from 30 hpf to 5 dpf. After rinsing, DAF-FM DA staining was performed for 3 h, followed by live imaging, as described above. Controls of the inhibition were carried out using DMSO at a similar concentration as the inhibitor (0.1%).

### 2.5. Processing for GMA Sections

All specimens live-stained with DAF-FM DA, as well as their controls, were embedded in GMA according to a published protocol [18]. Briefly, they were dehydrated shortly in acetone and transferred to the GMA monomer for 24 h. Specimens were next placed into the monomer with an added catalyst, and blocks were allowed to polymerize over two days. Serial 3 µm sections were made using a Prosan HM350 microtome. Sections were temporarily mounted with PBS or permanently with Vectashield containing DAPI. During all processing steps, samples, as well as the sections obtained from them, were protected from light.

### 2.6. Demonstration of Mineralized Tissues

For the detection of mineral salts, both Von Kossa and Alizarin red S staining were used. The selected sections of specimens previously live-stained with DAF-FM DA were processed for Von Kossa staining, according to [19]. Briefly, after rinsing in PBS and distilled H_2_O, sections were immersed for 45 min in 1% AgNO_3_ under UV light, rinsed, soaked for 5 min in 3% Na_2_S_2_O_3_, rinsed, soaked for 5 min in Von Gieson stain solution, followed by a short differentiation in 96% ethanol, and air-dried. For observation, sections were temporarily coverslipped with PBS, and the same areas, photographed prior to Von Kossa staining with the GFP filter for the DAF-FM DA signal, were photographed using brightfield illumination. Likewise, Alizarin red S staining (0.5% solution, pH 9, 1 min) was applied onto GMA sections of specimens live-stained with DAF-FM DA and observed with a rhodamine filter for the Alizarin red S signal. Overlay pictures were produced using Photoshop version 6.0.

### 2.7. Elastin Staining

WT zebrafish of 5 dpf were euthanized using an overdose of MS222, fixed in PFA, and processed for paraffin embedding according to standard procedures. Serial 5 µm cross sections were made using a Prosan HM350 microtome. Elastin was demonstrated using an adapted Verhoeff’s protocol (without Von Gieson counterstain) [20]. Briefly, after deparaffinizing and hydration, sections were stained for 30 min in Verhoeff’s iodine solution, rinsed, and differentiated in 2% FeCl_2_ for time intervals of 15 s, alternating with rinsing in distilled water, observation and microphotography.

### 2.8. Processing for Epon Sections and Transmission Electron Microscopy

Zebrafish embryos of 5 dpf, treated with TRIM, as well as control embryos, were euthanized using an overdose of MS222, fixed in a mixture of 1.5% glutaraldehyde and 1.5% paraformaldehyde (PG), postfixed in OsO_4_ and processed for epon embedding according to a published protocol [21]. Semithin (1 µm) sections were stained with Toluidine blue for 2 min (0.2% Toluidine Blue, 2% Na_2_B_4_O_7_), rinsed with water, air-dried, and mounted with DPX (Fluka, Buchs, Switzerland). Ultrathin sections (70 nm) were cut with a Reichert Ultracut E, were contrasted with uranyl acetate and lead citrate, and examined under a Jeol JEM 1010 transmission electron microscope (Jeol Ltd., Tokyo, Japan) operating at 60 kV. Microphotographs were taken with a Veleta camera (Emsis, Muenster, Germany).

### 2.9. Observations and Microphotography

Observations and microphotography of GMA and semithin epon sections were performed using a Zeiss Axio Imager Z1 compound microscope equipped with epifluorescence. Photographs were taken with an Axiocam 503 color camera (Carl Zeiss, Oberkochen, Germany).

## 3. Results

Soaking early postembryonic zebrafish in DAF-FM DA according to published protocols specifically stained different bone elements, which is consistent with previous reports [12,13,14] (Figure 1A,B). In addition, a strong signal was observed around the notochord and anterior to the heart, similar to what has been reported in the literature cited above. In contrast, cartilage gave a signal hardly detectable against the background. Although the irregular ventral border of the opercular bone may suggest the presence of labeled osteoblasts (Figure 1B), the signal associated with bony structures, as well as with the teeth, appeared, in sections, to be located in the extracellular matrix and not in the bone-forming cells, the osteoblasts (Figure 1C and Appendix A). Likewise, the signal associated with the notochord was located in the notochord sheath (Figure 1D); the signal associated with the heart was limited to the bulbus arteriosus and appeared to label intercellularly located fibrous material (Figure 1E).

We reasoned that three hours of staining might allow non-specific reactions to build up in the extracellular matrix, masking or perhaps abolishing an earlier intracellular signal. Thus, we soaked 5 dpf embryos in DAF-FM DA according to the recommended procedure but observed and sacrificed embryos every 20 min for the 3 h duration of staining (Figure 2). However, even after the short-term interval of 20 min, the staining pattern was comparable to that after 3 h, albeit slightly weaker: fluorescent signals were observed in the bulbus arteriosus of the heart, the notochord sheath, the matrix of the forming bones, and the dentine and attachment bone of the teeth (Figure 2 and Appendix A).

To exclude the possibility that the signal first appears inside the cells (e.g., osteoblasts) and subsequently leaks out into the ECM, we performed the same experiment, using 5 dpf embryos but exposing them for even shorter time intervals, with live observations after 1, 3, 5, 10 and 20 min. Under these conditions, the notochord sheath and the bulbus arteriosus were the first structures to give a signal after 3 min. A slight signal appeared in the opercular bone after 5 min. Although very weak after these short time intervals of staining, the signal in the bones was again clearly limited to the matrix (Appendix A).

To assess whether the staining pattern observed could simply be an artifact unrelated to the live uptake of the compound, we also performed DAF-FM DA staining on fixed material. The three different fixatives used (4% PFA, acetone, or 70% ethanol) showed different patterns of fluorescence, with PFA being the strongest, including a dotted pattern in the skin as well as a signal in axial structures, but not the notochord. Control zebrafish that were treated with DMSO showed hardly any fluorescence. This shows that staining after fixation can indeed yield a specific pattern of fluorescence, which is, however, different from that obtained after live staining (Appendix A).

Next, we compared live DAF-FM DA staining with or without pretreatment with the NO scavenger c-PTIO (Figure 3). Staining with DAF-FM DA after treatment with the NO scavenger c-PTIO weakened but did not abolish the signal.

Because staining with DAF-FM DA appeared to be associated most with mineralized structures (bones and teeth–the notochord sheath mineralizes later), we asked whether the mineral in these structures could cause non-specific staining, i.e., whether the fluorescent signal could coincide with mineralized areas. To address this question, we selected a number of sections with a strong fluorescent signal from the live staining experiment and reutilized these for Von Kossa staining. The overlay of the fluorescent DAF-FM DA and brightfield Von Kossa images clearly showed more extensive fluorescent staining than that revealed by Von Kossa staining for mineralized areas (Figure 4). Thus, some (mostly smaller, thinner) bones that were clearly stained using DAF-FM DA did not show any staining for minerals (Figure 4A–A″). Other (larger) bones consistently showed a rim of fluorescent staining that was negative with Von Kossa, likely corresponding to the osteoid layer of the bone (Figure 4C–C″). Staining with Alizarin red S exactly replicated the results obtained with Von Kossa staining: DAF-FM DA labeled small bones that were left unstained with Alizarin red S (i.e., not yet mineralized) (Figure 4B–B″), as well as non-mineralized and mineralized (Alizarin red S-positive) areas of larger bones (Figure 4D–D″). This indicates that DAF-FM DA stains the organic extracellular matrix and not its mineral component.

To assess whether the signal in the bulbus arteriosus (Figure 5A) corresponds to elastin, we compared the distribution of the signal with elastin, as revealed by Verhoeff’s elastin stain. Since Verhoeff’s staining cannot be performed on GMA-embedded sections, we necessarily had to compare images from different animals. However, abundant and distinct elastic fibers were demonstrated in the bulbus arteriosus (Figure 5C), which were similar in distribution and morphology to the fluorescent signals observed after DAF-FM DA staining (Figure 5B). Surprisingly, TEM pictures of the 5 dpf zebrafish bulbus (Figure 5D) revealed only weak evidence of elastin fibers, suggesting that the fluorescent signal may reveal components associated with the elastin-rich extracellular matrix.

NO is assumed to play a role in bone cell metabolism [22]. To assess whether the surprising absence of staining in the osteoblasts is a failure due to technical or chemical limitations, we applied TRIM, an inhibitor of all NOS enzymes that catalyze the production of NO, and subsequently stained the treated embryos with DAF-FM DA. Live observations from TRIM-treated and DAF-FM DA-stained embryos (Figure 6A,B), as well as whole mount staining (Figure 6C,D) and sections prepared from TRIM-treated embryos (Figure 6E–H′), clearly revealed that bone structures (opercular, cleithrum) were absent (compare Figure 6E,E′ with Figure 6G,G′ for the opercular, and Figure 6F,F′ with Figure 6H,H′ for the cleithrum).

This result supports the implication of NO production in bone formation and challenges DAF-FM DA as a marker of intracellular NO production in osteoblasts.

## 4. Discussion

A clear and consistent result of live staining zebrafish embryos with DAF-FM DA is the strong signal in the extracellular matrix, whether from the bone or from teeth (dentine, bone of attachment). The lack of signal in bone-related cells is especially surprising, given the regulatory role of NO in bone formation [5,6,7,8,9,22], as supported by our inhibition experiments. To explain this rather puzzling result, we are hindered by the fact that staining must be carried out on live specimens. This limits the use of relevant controls prior to or during staining, such as decalcification or enzymatic digestion of extracellular components. On the other hand, the results may not be so surprising given that the vast majority of studies that have utilized DAF-FM DA to demonstrate NO have been performed on isolated cells in culture or on organ fragments, i.e., in vitro or ex vivo (e.g., [23,24,25]).

A few studies have reported on the live staining of early postembryonic zebrafish using DAF-FM DA. Although some of these papers used DAF-FM DA specifically for skeletal studies [13,14], they mostly refrained from attributing the signals directly to osteoblasts and referred to the signals as issuing from ‘forming bones’ [12] or ‘ossified structures’ [13].

Here, we show that the compound, which was developed to reveal intracellular NO [11], is exclusively labeling the extracellular matrix. Renn et al. [13] noticed already that, at 6 dpf, the signal labeled the opercular bone but not the osteoblasts lining the bone and attributed this to an altered NO production status of the cells. In the present study, we show that the lack of DAF-FM DA signal in the osteoblasts is not limited to later postembryonic stages; rather, it is the matrix that is labeled throughout early postembryonic development. Our experiments targeting the inhibition of NO production using TRIM nevertheless clearly show the absence of bone structures and, thus, are a strong indication of the role of NO in bone formation.

Different explanations can be proposed for the lack of intracellular staining of the osteoblasts. One possibility is that in an in vivo context, NO diffuses out of osteoblasts within seconds after its production and reacts with DAF-FM DA even before this is able to penetrate into the cells. Under physiological conditions, NO, as an uncharged molecule, is readily diffusible, with a range of approximately 150–300 µm for a time of 4–15 s. This distance does not correspond to a straight-line trajectory (since the process is random) but rather the radius of a sphere [26]. Under certain conditions, the half-life of NO can reach 10 min in a solution [27]. The above scenario raises the question of where the conversion occurs into the triazole form (which is necessary to produce a strongly fluorescent molecule); does this occur in the extracellular space or within the bone matrix itself? The second option requires DAF-FM DA to have spread within the matrix before NO. In a recent study, where anosteocytic (medaka) and osteocytic (zebrafish) bone was compared, it was observed that water can easily diffuse and exchange within the extracellular matrix in teleost bones, and more so in medaka than in zebrafish [28]. These results are relevant since the bones in early postembryonic zebrafish are virtually anosteocytic [29], making them more similar to medaka bone. Thus, one may hypothesize an easy access of intercollagenous spaces for diffusion. Still, this scenario also requires that the diacetate in DAF-FM DA does not interfere with the formation of a triazole. The observation that no staining is ever observed in the extracellular space surrounding the bone suggests that the reactions turning DAF-FM DA into a fluorescent derivative take place within the matrix itself.

The weak effect on staining after the use of the NO scavenger c-PTIO, in contrast with the dramatic loss of staining after the use of the NOS inhibitor TRIM, can be explained by the different chemical actions of both compounds: c-PTIO oxidizes the NO that has been produced; TRIM blocks the enzymes necessary for NO production. In addition, many pitfalls have been identified regarding the use of c-PTIO, at least in plants [30]. For example, Arita et al. [31] showed that, contrary to the widely held presumption, the NO scavenger c-PTIO does not suppress but actually enhances the conversion of DAF into the DAF-2T fluorophore.

It is important to point out that the conversion from DAF-FM to its triazole product is probably not direct. DAF-FM is weakly fluorescent and is likely to be first non-specifically oxidized to an anilinyl radical, which then reacts with NO to form the fluorescent triazole product [32]. It was reported that DAF-2 rapidly reacts with dehydroascorbic acid (DHA) under physiologically relevant conditions to generate compounds (DAF-2-DHAs) that have fluorescence emission profiles similar to that of DAF-2 triazole (DAF-2T) [33]. Likewise, Wardman [32] describes how DAF-2 can be non-specifically oxidized to yield a fluorescent molecule. Whether the same is true for DAF-FM, which is a DAF-4 derivative, is unclear to us. Moreover, according to sources cited in [34], DAF-2 reacts with peroxynitrite rather than nitric oxide.

An entirely different explanation for the fluorescent signal in the bone and dentine matrix may relate to their calcium content. It has been both reported and rebutted that DAF-2 is sensitive to the presence of divalent cations, especially calcium, in the medium [34]. However, our observations that DAF-FM DA stains small, not-yet mineralized bones, as well as the osteoid layer around mineralized bones, argue against the calcium sensitivity of DAF-FM DA. Likewise, Renn et al. [13] observed that DAF-FM DA labels the non-mineralized osteoid of the early cleithrum.

In the bulbus arteriosus, the staining is also extracellular, as revealed by DAPI staining, and is, thus, excluded both from endothelial and smooth muscle cells. Instead, the intensive staining with DAF-FM DA matches the distribution of elastic fibers, as demonstrated by Verhoeff’s staining, although direct superimposition with the fluorescent signal is not possible using this technique. Ultrastructural observations suggest that the signal might derive from elastin-associated material rather than from elastin fibers proper. The bulbus arteriosus in teleosts is known to have a multilayered structure, with the largest, middle layer containing abundant elastic fibers [35,36]. In sticklebacks, the elastic fibers are reported to be 15 nm in diameter, associated with amorphous material and completely fill the space between the adjacent rows of smooth muscle cells [35]. Interestingly, Rodriguez et al. [37] already reported the strong staining of rat aortic tissue with DAF-2T DA. Similar to our conclusion that the signal is located in the elastic fiber-rich extracellular space, they suggested that the emission of DAF-2 T from aortic tissue originates predominantly from the elastic laminae. They also proposed several mechanisms that could account for the localization of this signal, drawn from earlier studies, such as enhanced N_2_O_3_ production occurring within the hydrophobic environment of the lamina [37]. Alternatively, it may reflect the preferential accumulation of the fluorophore within elastic fibers. This explanation is supported by the reported interaction between fluorescein and collagen, suggesting that this type of fluorophore binds strongly to these proteins [37]. According to McCarthy [38], reactive dyes such as the succinimidyl ester of carboxyfluorescein diacetate [CFDA] bind covalently and predominantly to proteins.

Finally, sections show that in the notochord, the NO signal is clearly limited to the notochord sheath and does not label the cytoplasm of the vacuolated cells [12]. The notochord sheath is an extracellular matrix containing collagens and elastin (reviewed in [39]).

Interestingly, the bone and dentine matrix, notochord sheath, and bulbus elastic laminae are not the only extracellular compartments that supposedly reveal the presence of NO. NO is also found in saliva, plasma, and blood [40]. Lundberg et al. [41] observed elevated NO levels in the gastric lumen, probably resulting, in the authors’ view, from non-enzymatic NO production and requiring an acidic environment. The latter was justified by the observation that NO in expelled air was reduced by 95% after pretreatment with the proton pump inhibitor omeprazole. The substrate for intragastric NO formation is probably nitrite, given that nitrite is reduced in an acidic environment, thus forming NO [42]. More recently, Wang et al. [43] confirmed that the reaction of DAF-FM DA with nitrite in an acidic medium results in the formation of triazolofluorescein (DAF-FM T). A reaction with nitrite in the bone matrix of zebrafish embryos is nevertheless unlikely based on the pH of bone matrix (for rodent calvariae, around pH 7.4 [44]) as well as on the toxicity of nitrite. Moreover, NO would still be the most likely source of nitrite.

## 5. Conclusions

In conclusion, the fluorescence patterns observed when using DAF-FM DA as a live stain for zebrafish embryos may not necessarily correlate with local NO production, nor do these patterns exclude the concept that cells responsible for the deposition of this matrix can produce NO. Indeed, the inhibition of NO production at early embryonic stages completely abolishes all bone formation, suggesting an important role for NO in osteoblast function. However, DAF-FM DA is not the appropriate technique to demonstrate NO production in osteoblasts, chordoblasts, or smooth muscle cells in the in vivo context of zebrafish embryos. Additional controls must be carried out; not least, the analysis of these signals should be performed at a sufficient resolution to assert whether the signal is found intra- or extracellularly. Irrespective of these caveats, the compound can be used in zebrafish research to visualize specific anatomical structures, including the notochord, the bone matrix in early ossification centers, the teeth, and the elastic fibers in the bulbus arteriosus. Because many studies that use zebrafish to model human bone diseases rely on demonstrating mineralized structures with Alizarin red S, DAF-FM DA staining can become an easy-to-perform and quick evaluation method to assess the presence of the earliest osteoid. Likewise, DAF-FM DA staining could be a valuable tool for the live imaging of heart function.

## Figures and Tables

**Figure 1 biomolecules-13-01780-f001:**
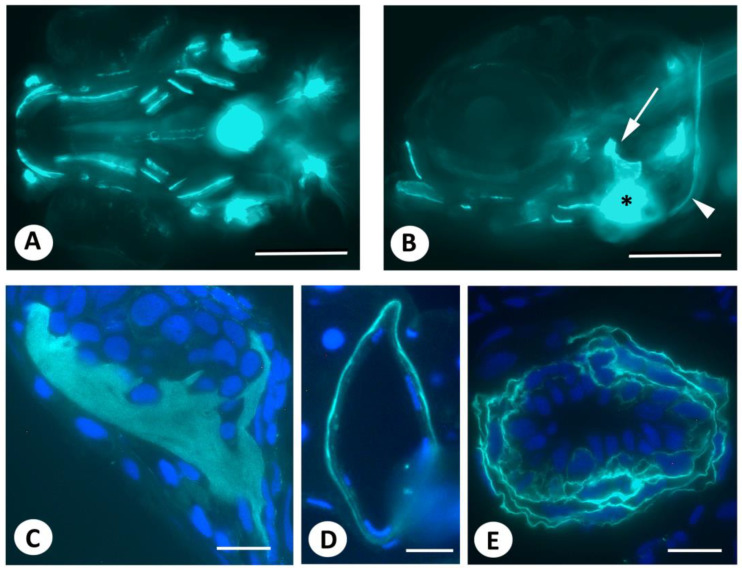
Live staining of early postembryonic zebrafish embryos with DAF-FM DA. (**A**,**B**). Live staining with DAF-FM DA, according to published protocols, reveals signals in bony structures (opercular, arrow; cleithrum, arrowhead), as well as in the heart (asterisk) and around the notochord, which is consistent with earlier reports. Axio Observer image. (**C**–**E**). Three µm GMA sections of the specimen shown in (**A**,**B**), counterstained with DAPI, showing the opercular bone (**C**), notochord (**D**), and bulbus arteriosus (**E**). Note that nuclei lie outside fluorescent domains. Scale bars in (**A**,**B**) = 200 µm, in (**C**,**E**) = 20 µm, in (**D**) = 10 µm.

**Figure 2 biomolecules-13-01780-f002:**
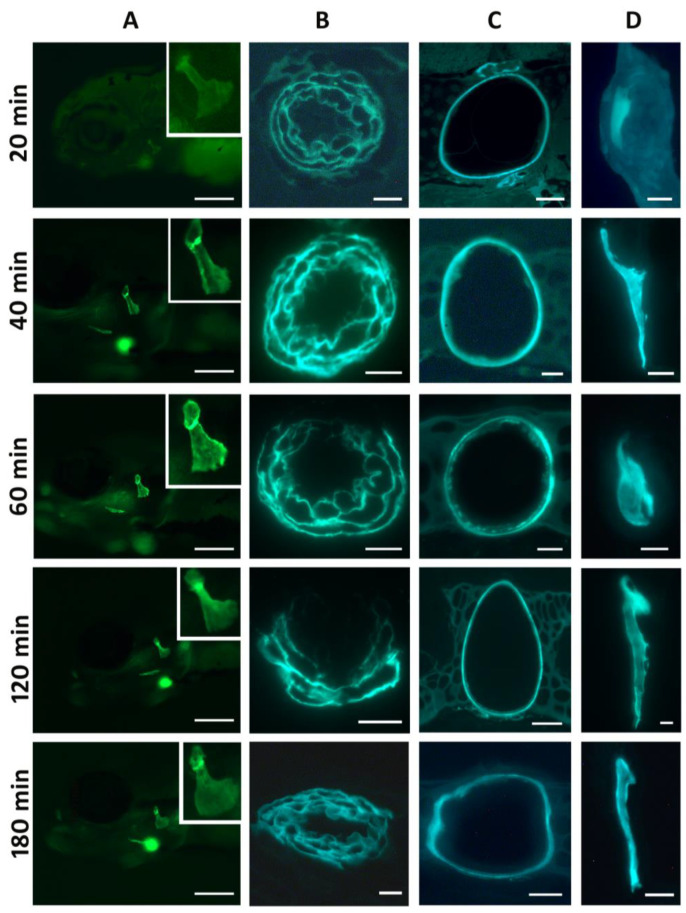
Live staining of 5 dpf zebrafish embryos with DAF-FM DA for short intervals. Column (**A**): live imaging of the head region at the time indicated on the left; inset: magnification of the opercular bone. The strong blurred signal visible ventrally is the bulbus arteriosus. Scale bars = 200 µm. Columns (**B**–**D**): 3 µm GMA sections of the corresponding embryos. Column (**B**): bulbus arteriosus. Scale bars = 10 µm. (**C**): notochord in the head region. Scale bars = 20 µm except for 60 min: 10 µm. (**D**): opercular bone. Scale bars = 10 µm except for 40 min: 20 µm. Column (**A**): anterior to the left; (**B**,**C**): dorsal to the top; (**D**): lateral to the left.

**Figure 3 biomolecules-13-01780-f003:**
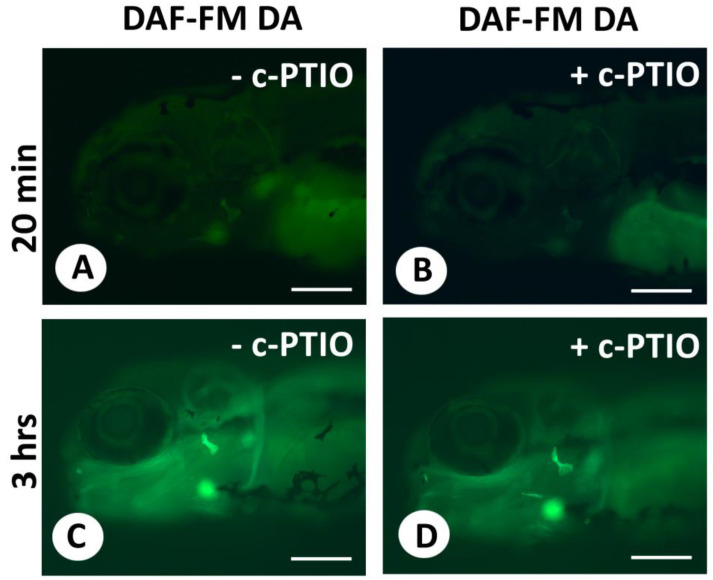
Live staining of 5 dpf zebrafish with DAF-FM DA after the use of the NO scavenger c-PTIO. (**A**–**D**). Live staining of 5 dpf zebrafish for 20 min (**A**,**B**) or 3 h (**C**,**D**) with DAF-FM DA either without c-PTIO (**A**,**C**) or after 30 min treatment with the NO scavenger c-PTIO (**B**,**D**). All images were taken strictly under the same illumination. Scale bars = 200 µm.

**Figure 4 biomolecules-13-01780-f004:**
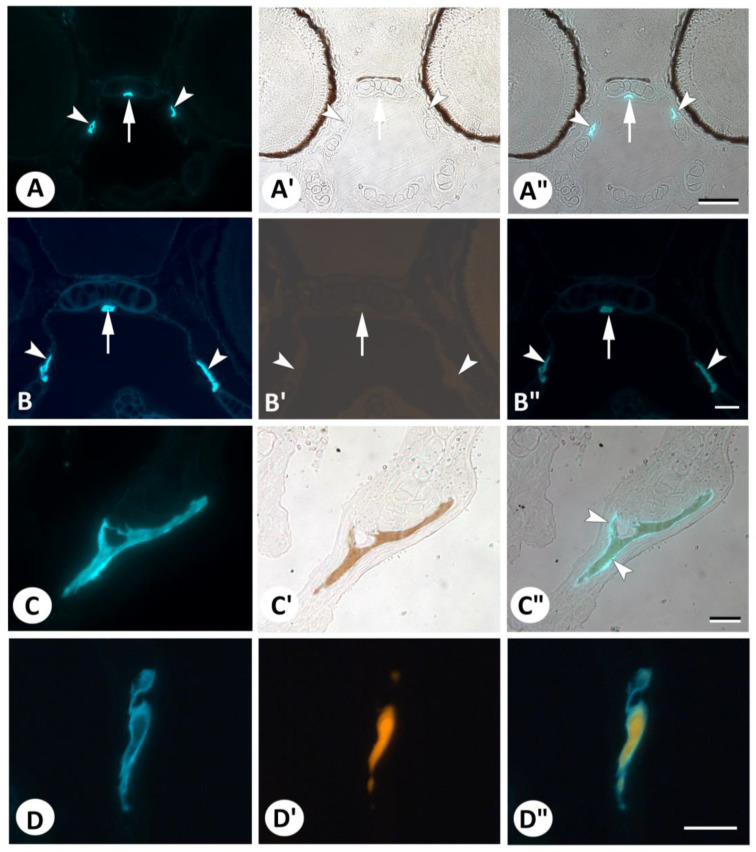
Comparison of DAF-FM DA-stained structures with staining for mineralized tissue. (**A**–**A″**). Overview section of DAF-FM DA live-stained 5 dpf zebrafish (**A**) stained with Von Kossa for mineralized structures (**A′**) and overlay (**A″**). Note the absence of Von Kossa staining in the para-sphenoid (arrow) and entopterygoid bones (arrowheads). (**B**–**B″**). Overview section of a DAF-FM DA live-stained 5 dpf zebrafish (**B**) stained with Alizarin red S for mineralized structures (**B′**) and overlay (**B″**). The image is dark because of the complete absence of Alizarin red S staining in the parasphenoid (arrow) and entopterygoid bones (arrowheads) (compared with the positive staining of mineralized bone in D′). (**C**–**C″**). Details of the opercular bone in a section of a DAF-FM DA live-stained 5 dpf zebrafish (**C**) stained with Von Kossa for mineralized structures (**C′**) and overlay (**C″**). Note the distinct zone of DAF-FM DA-positive staining around the area positive for minerals using Von Kossa (arrowheads). (**D**–**D″**). Details of the opercular bone in a section of a DAF-FM DA live-stained 5 dpf zebrafish (**D**) stained with Alizarin red S for mineralized structures (**D′**) and overlay (**D″**). Note that the DAF-FM DA-positive area is clearly larger than the area marked with Alizarin red S. Scale bar for (**A**–**A″**) = 50 µm and for (**B**–**B″**) to (**D**–**D″**) = 20 µm.

**Figure 5 biomolecules-13-01780-f005:**
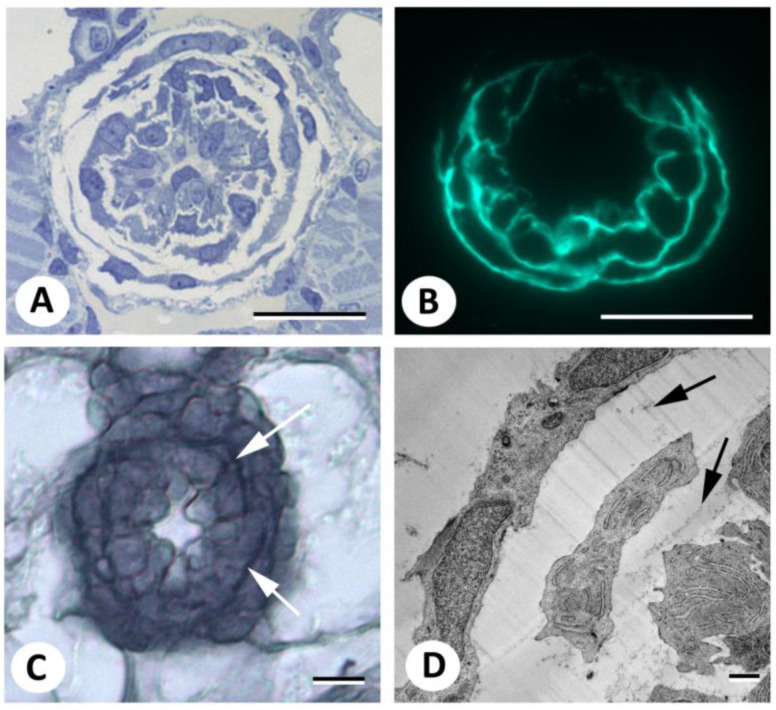
Demonstration of elastic fibers in the bulbus arteriosus. (**A**). Semithin Toluidine blue-stained cross section of the bulbus arteriosus in a 5 dpf zebrafish. (**B**). Section of the bulbus after live staining of a 5 dpf zebrafish with DAF-FM DA. (**C**). Elastin staining of the bulbus according to an adapted Verhoeff’s stain. (**D**). TEM image of the bulbus of a 5 dpf zebrafish. Arrows indicate the ECM (elastin fibers in (**C**)). Scale bars in (**A**–**C**) = 20 µm, in (**D**) = 1 µm.

**Figure 6 biomolecules-13-01780-f006:**
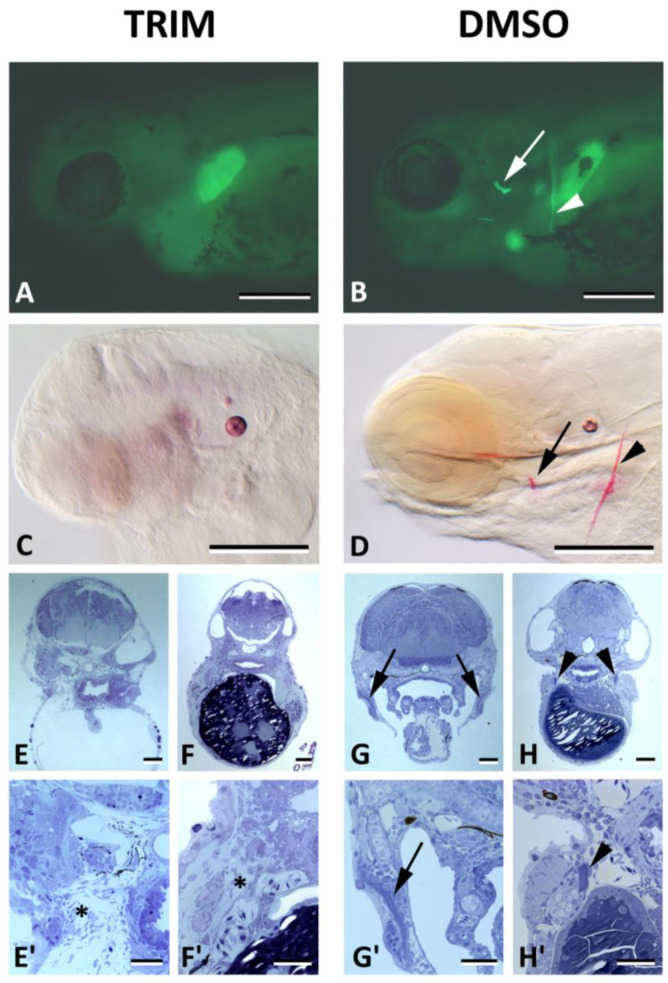
Absence of bone after inhibition of NO formation. (**A**,**B**). Live staining of 5 dpf zebrafish with DAF-FM DA for 3 h in the dark after 90 h of treatment with the NOS inhibitor TRIM (**A**) or its solvens (0.1% DMSO) (**B**). Note the presence of opercular bone (arrow) and cleithrum (arrowhead) in the control, which are clearly absent in the TRIM-treated specimen. (**C**,**D**). Alizarin red S whole mount staining of TRIM-treated embryos likewise shows the complete absence of bones (**C**) compared to their presence in control embryos (**D**). Arrow: opercular bone; arrowhead: cleithrum. (**E**–**H′**). Semithin sections at comparative cross-sectional levels show the absence of opercular bone and cleithrum ((**E**,**F**), and higher magnification in (**E′**,**F′**)) compared to their distinct presence in control embryos ((**G**,**H**), and their higher magnification in (**G′**,**H′**)). Opercular bone: arrow; cleithrum: arrowhead; asterisks indicate the location of absent bones. Scale bars in (**A**,**B**) = 200 µm, in (**E**–**H**) = 100 µm, in (**E′**–**H′**) = 50 µm.

## Data Availability

The sections pictured in this paper are kept in the slide collection of the Research Group Evolutionary Developmental Biology of Ghent University and are available for inspection upon request.

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
