# Peer review of "Bone Formation in Zebrafish: The Significance of DAF-FM DA Staining for Nitric Oxide Detection"

_biomolecules, 2023, doi:10.3390/biom13121780_

Round 1

Reviewer 1 Report

Comments and Suggestions for Authors

This article presents the use of DAF-FM DA staining as a technical approach to visualize the developing skeletal elements in zebrafish by marking its extracellular matrix. Although, and as pointed by the authors, this reagent is not specifically reacting with NO in the studied tissues, and the chemical reaction leading to the marking is still not understood, still this can be a useful technique for studying the forming extracellular matrix even before mineralization starts to occur. It can be of particular interest for studies of conditions where the ECM mineralization is compromised, not only in the context of fish skeleton development, but also to access human conditions like nutritional induced rickets and osteomalacia. 

Some points in the methods deserve better explanation, like the rationale for performing the experiment with treatment from 1 to 20 min. Authors shown that 20 min exposure already show a weaker signal, so why going to even lower exposure times?

Also in lines 86-87 it is mentioned that in specific cases C-PTIO was employed. As a method description this is not clear and replicability is not ensured, therefore authors should clearly explain in which specific cases. The full name of the molecule should also be provided. 

In the results and discussion is mentioned that dentine and attachment bone for teeth is marked, but as detailed in figures 1 and 2 it is not clear the marking in these structures. The authors must clearly indicate where is the signal is located in the images presented. 

Specific Comments:

Lines 217-218 The authors state that "staining with DAF-FM DA appeared to be associated with mineralized structures (bones, teeth)", however the elastic arterial bulbus and the notochord sheath are not mineralized structures and are also marked, therefore unspecific staining due to mineral should not be the cause.

Figure 5- the image n fig 5B is the same image as shown in figure 2 at 40 min. In fig 5D the elastin fibers should be clearly indicated

Lines 276-277 In addition these TRIM treated images also show that the arterial bulbous show no signal.  Was it also not formed or is it absent? Or it has to do with ablation of NO production? Please comment

Line 283 Alizarine should be Alizarin

The discussion is very focused on the chemical aspects of DAF-FM and some more discussion on the applications that can be made using the technique described would be interesting to the broader audience.

Reviewer 2 Report

Comments and Suggestions for Authors

In this work, Huysseune et al., have used the chemical compound DAF-FM DA to detect nitric oxide in live zebrafish. Staining includes bone structures such as the notochord, operculum, and various jaw elements. While this has been demonstrated previously, this work adds to the body of literature by showing the signal is likely coming from the bone matrix as opposed to osteoblasts themselves.  While the work is generally well done, there are number of concerns that need to be address before publication.

Major:

1)    The nitrogen scavenging compound (PICO) results are not convincing. This is a very subtle reduction in fluorescent intensity when incubated in this compound, in contrast to the much more severe reduction in staining when nitrogen oxide synthase is inhibited with TRIM. This discrepancy needs to be addressed in the discussion. Ideally, a dose response and time course utilizing the PICO compound would be tested, as different concentrations and time of staining may yield better results. 

2)    The DAF-FM staining in Medaka (supplementary figure 3) is very mild and not all that convincing. Although there is some signal in the bulbus arteriosus, overall staining in this structure and jaw elements is much weaker that in zebrafish. At least, a control (DMSO) should be included to ensure this staining is specific. 

3)    Many of the panels in figures are not referred to in the results section. For example, (lines 274-278) states “Live observations from TRIM-treated and DAF-FM DA stained embryos, as well as whole mount staining and sections prepared from TRIM-treated embryos, clearly revealed that bone structures (opercular, cleithrum) were absent (Figure 6). None of the individual panels, that comprise three different techniques, are specifically referred to in the results text. This is also the case for figures 5.

4)    The authors contradict themselves at times, with respect to figure 4 (lines 228-232) “DAF-FM DA labeled small bones that were left unstained with Alizarin red S (i.e., not yet mineralized) (Figure 4B-B”), as well as non-mineralized and mineralized (Alizarin red S-positive) areas of larger bones (Figure 4D- D”). This indicates that DAF-FM DA stains the organic extracellular matrix, and not its mineral component. 

5)    It is difficult to see anything in figure 4B’. This picture should be replaced.

6)    In figure 6, panel C appears to be a lateral view, while panel D is a ventral view. It is difficult to compare such pictures.

Minor. 

1)    It is not stated what the scale bar is in figure 5D. It is not clear what in panel 5D is an elastin fiber. This should be highlighted with an arrow.

2)    In the figure legend for S1, the bulbus arteriosus is incorrectly referred to as the bulbus olfactorius.

3)    Line 218. “unspecific staining” should be replaced with “non-specific staining.”

4)    Line 282- the phrase “completely similar” is itself a contradiction. This should be replaced. 

5)    Can the authors highlight in Figure 1 where the teeth are?

Comments on the Quality of English Language

Minor editing required.
